# Activatable probes for diagnosing and positioning liver injury and metastatic tumors by multispectral optoacoustic tomography

Yinglong Wu [1], Shuailing Huang[1], Jun Wang[1], Lihe Sun [1], Fang Zeng[1] & Shuizhu Wu [1]

Optoacoustic tomography (photoacoustic tomography) is an emerging imaging technology displaying great potential for medical diagnosis and preclinical research. Rationally designing activatable optoacoustic probes capable of diagnosing diseases and locating their foci can bring into full play the role of optoacoustic tomography (OAT) as a promising noninvasive imaging modality. Here we report two xanthene-based optoacoustic probes ($C^1X$-$OR^1$ and $C^2X$-$OR^2$) for temporospatial imaging of hepatic alkaline phosphatase (or β-galactosidase) for evaluating and locating drug-induced liver injury (or metastatic tumor). The probes rapidly respond to the disease-specific biomarkers by displaying red-shifted NIR absorption bands and generate prominent optoacoustic signals. Using multispectral optoacoustic tomography (MSOT), we can precisely localize the focus of drug-induced liver injury in mice using $C^1X$-$OR^1$, and the metastatic tumors using $C^2X$-$OR^2$. This work suggests that the activatable optoacoustic chromophores may potentially be applied for diagnosing and localizing disease foci, especially smaller and deeper ones.

[1] State Key Laboratory of Luminescent Materials & Devices, College of Materials Science & Engineering, South China University of Technology, Guangzhou 510640, China. Correspondence and requests for materials should be addressed to F.Z. (email: mcfzeng@scut.edu.cn) or to S.W. (email: shzhwu@scut.edu.cn)

For visualizing and quantifying biological processes at molecular levels, noninvasive imaging is an advantageous approach to diagnose, predict, stage, and monitor the development of diseases. As a noninvasive optical imaging modality, fluorescence imaging has been widely employed and provided valuable information for medical diagnosis and preclinical research[1–5]. However, the strong light scattering in tissue causes the spatial resolution of the fluorescent signal to degrade rapidly with imaging depth. On the other hand, by adding ultrasound detection to optical excitation, optoacoustic tomography (OAT), also known as photoacoustic tomography (PAT), has emerged as a promising imaging modality through detecting the ultrasound waves generated by the thermoelastic expansion of tissue as a result of laser pulse absorption[6–16]. In particular, multispectral optoacoustic tomography (MSOT)[8–10], which is a spectral optoacoustic technique, has been utilized in a wide range of biological imaging applications[17–20]. A MSOT system operates by irradiating a sample with multiple wavelengths, allowing it to detect ultrasound waves from different photoabsorbing substances in the tissue. Afterwards, computational techniques, such as spectral unmixing, deconvolute the ultrasound waves emitted by these different absorbers, allowing each photoabsorber to be visualized separately in the target tissue. In this way, MSOT can distinguish ultrasound signals of exogenous contrast agents from the background signals of hemoglobin, melanin, and etc. Moreover, three-dimensional (3D) MSOT images can be obtained by volumetric imaging technique, or by rendering stacks of 2D images as 3D images[8,10].

To date, some exogenous optoacoustic contrast agents like organic dyes[21–28], carbon nanomaterials[29–32], metal nanoparticles[33–41], and etc. have been developed for this fast-growing imaging technology for tumor detection[42–46], therapeutic monitoring[47–52], reactive oxygen species imaging[53], metal ion indication[54], and so on. For the contrast agents, the activatable ones capable of responding to specific biological stimuli and generating strong optoacoustic signals are particularly desirable, because they can achieve high sensitivity detection and allow for real-time tracking of dynamic processes[55]. However, there are very limited reports so far on the design and development of activatable optoacoustic contrast agents, especially the small molecular ones[23,24,52], let alone using them for precisely positioning diseases via obtaining MSOT images with 3D information.

In medicine, biomarkers are measurable indicators of the severity or presence of some disease state; they encompass a wide variety of molecules, such as enzymes, metabolites, nucleic acids, and etc[56]. They are often assayed and evaluated for diagnosing specific diseases, in which specific biomarkers are consistently presented at abnormal concentrations. Currently, serum assay remains the mainstream approach for biomarker detection. However, many biomarkers reside in multiple organs or tissues besides the disease focus, which can compromise the detection specificity of serum assay[57]. For example, elevation in serum ALP is usually regarded to be associated with liver disorders; however, ALP is present in several tissues and organs including liver, bone, intestine, and placenta[58], thus the ALP elevation in serum does not necessarily mean the liver dysfunction; and only alkaline phosphatase (ALP) of hepatic origin can serve as an important indicator for liver disorders and damages[59,60]. Hence, using 3D rendering images, one can spatially localize the elevation of the biomarker level at the specific organ or tissue by using activatable optoacoustic probe, thereby greatly reducing the risk of false-positive signals.

Historically, a great number of molecular chromophores have been designed as the fluorescent and colorimetric sensors/probes for disease diagnosis by adopting some well-established photophysical and photochemical protocols[61]. These protocols may also be exploited to fabricate activatable molecular OA sensors for disease detection in vivo.

Herein, we report the design and construction of near infrared (NIR) activatable chromophores as the optoacoustic/fluorescent dual-mode turn-on imaging systems for disease-specific biomarkers detection and imaging. As a proof of concept, we prepared two xanthene derivatives for diseases diagnosis and tracking; and we used these probes to diagnose and monitor (a) the drug-induced liver injury and subsequent rehabilitation by imaging hepatic ALP activity, and (b) the metastasized tumors of ovarian cancer in abdominal cavity and lymphatic metastasis by imaging β-galactosidase (Gal) level in mouse model. The schematic illustration for the detection mechanisms is shown in Fig. 1a. Our results indicate that, the probes can quickly respond to the activity change of the corresponding disease biomarkers and thereby provide temporal and 3D spatial information of the disease foci.

## Results

**Synthesis of activatable molecular OA probes**. To synthesize the two activatable OA probes ($C^1X$-$OR^1$ and $C^2X$-$OR^2$), we first obtained two NIR chromophores ($C^1X$-OH and $C^2X$-OH, they are also the activated form of the probes) by respectively incorporating different indole derivatives onto the xanthene structure, and then attached two recognition elements ($R^1$ and $R^2$) onto the respective chromophores (Supplementary Figure 1). The structures of the two probes are given in Fig. 1a. The probe $C^1X$-$OR^1$ was utilized to image drug-induced liver injury via responding to hepatic ALP activity; while the molecular probe $C^2X$-$OR^2$ was designed to image metastatic tumors via sensing the activity of β-galactosidase. The probes were synthesized according to the routes shown in Supplementary Figure 1 and well-characterized by NMR and mass spectrometry (Supplementary Figure 2-19).

The sensing mechanisms are based on the change in intramolecular electronic push-pull states upon the biomarker-mediated chemical reactions. Under the catalysis of ALP, the electron-withdrawing phosphate ($R^1$) in the probe $C^1X$-$OR^1$ can be hydrolyzed to the electron-donating group hydroxyl, thereby turning the probe into its activated form ($C^1X$-OH) with a red-shifted absorption band at 684 nm, as shown in Fig. 1b. On the other hand, mediated by β-galactosidase, the probe $C^2X$-$OR^2$ can be hydrolyzed into $C^2X$-OH, which shows a red-shifted absorption band at 703 nm (Fig. 1c). These biomarker-mediated structural transformations cause the generation of both optoacoustic and fluorescent signal (Supplementary Figure 20–22), thereby realizing the dual-mode sensing/imaging of the biomarkers (ALP and β-galactosidase). In this study, we encapsulated the probe $C^1X$-$OR^1$ in phospholipid liposomes to enhance the solubility of the probes in blood. Furthermore, to afford the liver targeting capability for the liposomal $C^1X$-$OR^1$, a bile acid modified and hepatocyte-targeting phospholipid DSPE-PEG$_{2000}$-ChA (Fig. 1d) was used together with several other phospholipids during the liposome formation; and in in-vivo mouse model experiments the liposomal $C^1X$-$OR^1$ was administered by tail vein injection. A typical transmission electron microscopic (TEM) image and a liposome diameter distribution determined by dynamic light scattering are presented in Fig. 1e, f respectively, which indicate that the average hydrodynamic diameter of the liposomal $C^1X$-$OR^1$ is about 105 nm.

**Spectral response of probes toward corresponding biomarkers.** Optoacoustic effect is based on the conversion of absorbed optical energy to heat and then to ultrasound waves, hence a probe's optoacoustic performance is closely related to its absorption properties and the quantum yields of the heat conversion-related

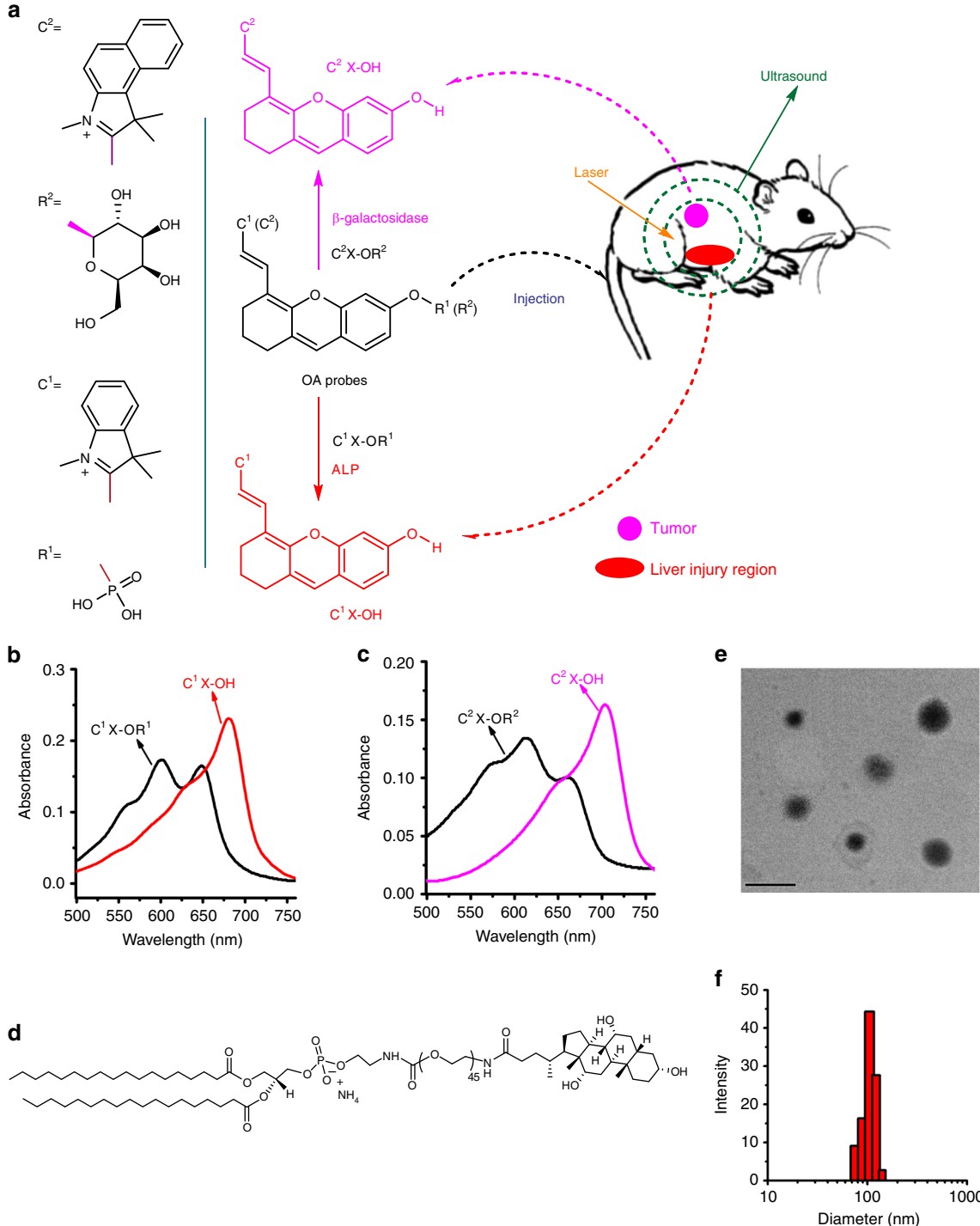

**Fig. 1** Schematic illustration for the probes' response in vivo and related properties. **a** Schematic illustration for two xanthene-based molecules ($C^1X\text{-}OR^1$ and $C^2X\text{-}OR^2$) as the activatable OA probes for respectively imaging liver injury and metastatic tumor. **b** and **c** Absorption spectra for the probes (5 μM) and their corresponding activated forms ($C^1X\text{-}OH$ and $C^2X\text{-}OH$, 3 μM). **d** Structure of a hepatocyte-targeting phospholipid for liver targeting. **e** and **f** Transmission electron microscopic image and particle diameter distribution for a liposomal $C^1X\text{-}OR^1$ sample. Scale bar: 200 nm

transitions following absorption. The time-dependent spectral measurements were performed after incubating the probe with the corresponding enzyme at 37 °C in buffer solution for varied time. Based on the results given in Supplementary Figure 21–22, we choose 30 min as the incubation time for the subsequent experiments. The detailed spectral responses of $C^1X\text{-}OR^1$ (or $C^2X\text{-}OR^2$) towards ALP (or β-galactosidase) are presented in Fig. 2a–c (or Fig. 2f–h) respectively. In the absence of ALP,

$C^1X\text{-}OR^1$ exhibits a strong absorption band from 550 to 650 nm (Fig. 2a) as well as weak fluorescence at 712 nm (Fig. 2c). While upon addition of ALP, the absorption red-shifted, correspondingly optoacoustic signal at around 684 nm gradually enhanced; and the fluorescent emission at 712 nm also increased remarkably (Fig. 2c). These spectral changes were caused by the ALP-mediated structural transformation from $C^1X\text{-}OR^1$ to $C^1X\text{-}OH$. To study the sensing selectivity, various potential interfering

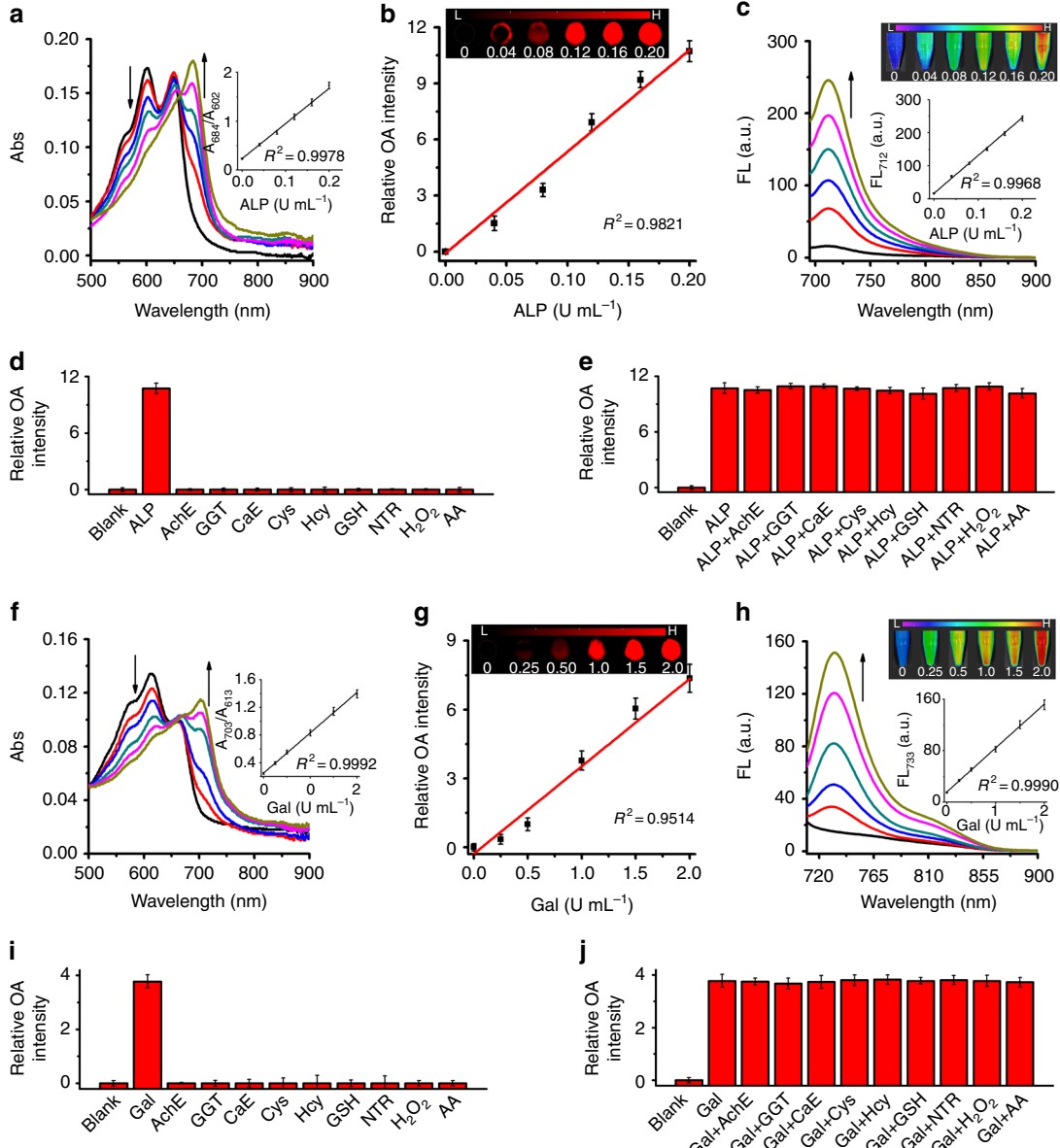

**Fig. 2** Optical responses of C$^1$X-OR$^1$ to ALP and C$^2$X-OR$^2$ to Gal in buffers. **a** and **f** Absorption spectra for the probes (5 μM) upon incubation with varied concentrations of corresponding biomarker (0–0.2 U mL$^{-1}$ for ALP and 0–2 U mL$^{-1}$ for Gal). The insets show the plots of absorbance ratio vs. biomarker concentration. **b** and **g** Optoacoustic response of the probes to corresponding biomarker of varied concentrations (ALP: 0–0.2 U mL$^{-1}$, Gal: 0–2 U mL$^{-1}$) ($n = 3$). The insets show the representative optoacoustic image of the probes in phantom at varied biomarker concentrations (excitation for **b** and **f** is 684 nm and 703 nm respectively). **c** and **h** Fluorescent spectra for the probes (5 μM) upon incubation with corresponding biomarker of varied concentrations (ALP: 0–0.2 U mL$^{-1}$, Gal: 0–2 U mL$^{-1}$). The insets show the plots of fluorescence intensity vs. biomarker concentration. **d** and **i** Optoacoustic response of the probes (5 μM) in the presence of 0.2 U mL$^{-1}$ ALP (1 U mL$^{-1}$ β-galactosidase) or a potential interference substance. **e** and **j** Optoacoustic response of the probes in the presence of ALP (β-galactosidase) and a potential interference substance ($n = 3$). Error bars represent the standard deviation (SD). Data were represented as mean ± SD

species were examined in parallel under the same conditions. As shown in Figs. 2d and e and Supplementary Figure 21c, C$^1$X-OR$^1$ shows high selectivity towards ALP over the other species tested. Following the same experimental protocol, we investigated the response of C$^2$X-OR$^2$ towards β-galactosidase. As can be seen in Fig. 2f–j, the probe C$^2$X-OR$^2$ also exhibited remarkable red-shift in absorption band as well as prominent fluorescence enhancement upon incubation with β-galactosidase. Moreover, the extinction coefficients for the two activated probes (C$^1$X-OH and C$^2$X-OH) were determined as $7.7 \times 10^4$ M$^{-1}$cm$^{-1}$ at 684 nm and

$5.4 \times 10^4$ M$^{-1}$cm$^{-1}$ at 703 nm respectively. These spectral results clearly indicate that the two probes can be potentially employed to image the corresponding biomarkers, both optoacoustically and fluorescently.

Moreover, the proposed response mechanism illustrated in Fig. 1a was verified by HPLC. As shown in Supplementary Figure 21d and 22d, for the solution containing C$^1$X-OR$^1$ (or C$^2$X-OR$^2$) only, a single peak at 7.3 min (or 5.8 min) can be observed. After incubation with ALP or β-galactosidase, a new peak at 12.5 min (or 10.1 min) appears, corresponding to the

activated probe $C^1X$-OH (or $C^2X$-OH) respectively. This verifies the transformation from the probes ($C^1X$-$OR^1$ and $C^2X$-$OR^2$) to the corresponding optoacoustically-active probes upon incubation with the biomarkers.

The sensitivities (the minimum amount of the activated probes necessary in order to be detectable) for the activated probes were determined using euthanized mice to evaluate the performance of the probes by MSOT imaging. As indicated in Supplementary Figure 23, the sensitivity limits for $C^1X$-OH and $C^2X$-OH were determined as 2.3 μM and 3.3 μM respectively, and the two probes are sensitive enough to image the relevant disease biomarkers.

**Fluorescence imaging of biomarkers by probes in live cells.** Prior to the cell imaging, the cytotoxicities of the probes were evaluated using several cell lines by MTT assay in compliance with ISO 10993-5. As shown in Supplementary Figure 24a and 25a, both probes show low cytotoxicity towards the cell lines. Notably, even after being incubated with the probe at the concentration of 50 μM, the viabilities of the cells are still higher than 85%. The low-toxic probes are suitable for imaging endogenous enzymes in live cells. Next, we investigated the imaging of intracellular enzymes by the two probes. Supplementary Figure 24b shows the time-dependent intracellular fluorescence for HepG2 cells (overexpressing ALP) upon incubation with $C^1X$-$OR^1$ by flow cytometry analysis. With the increasing incubation time, the fluorescence becomes more prominent, reflecting the occurrence of intracellular transformation from the probe $C^1X$-$OR^1$ to its activated form $C^1X$-OH as mediated by the hepatic ALP. The fluorescence response of $C^1X$-$OR^1$ to phosphatase was further confirmed by fluorescence microscope (Supplementary Fig. 24c). However, as for the L929 cells in which ALP is not over-expressed, no intracellular red fluorescence could be observed upon incubation 1 h with the probe (Supplementary Figure 24c). Similarly, $C^2X$-$OR^2$ was used to image β-galactosidase in three cell lines (L929, SHIN3, and OVCAR3). As can be seen in Supplementary Figure 25, in β-galactosidase over-expressing cells such as SHIN3 and OVCAR3, strong red fluorescence could be observed; while there was no fluorescence in L929 cells. There results indicate that the intracellular fluorescence enhancement is indeed caused by the endogenous over-expressed ALP or β-galactosidase.

**MSOT imaging of drug-induced liver injury in mouse model.** The probes exhibit low toxicity in vivo, which was confirmed by biochemical assay of serum AST, ALT, and ALP levels one or seven day(s) upon tail vein injection of liposomal probe $C^1X$-$OR^1$ and intraperitoneal injection of the molecular probe $C^2X$-$OR^2$, as well as the body weight changes and histological evaluation of major organs (Supplementary Figure 26 and 27). Next, N-acetyl-p-aminophenol (APAP), which is a commonly-used clinical drug for treating pain and fever and is well-known for its hepatotoxicity if used inappropriately, was utilized to induce liver injury in this study. APAP overdose increases elevated expression of ALP in liver[58,59]. Toxic dose of APAP was intraperitoneally administered to male nude mice, followed by i.v. injection of liposomal $C^1X$-$OR^1$ (with the net $C^1X$-$OR^1$ dosage of 3.7 mg kg$^{-1}$). The MSOT technology was then employed to detect the drug-induced liver injury through imaging the change in activity of the hepatic phosphatase, and the results are depicted in Fig. 3 and Supplementary Figure 28–31. Also, an example of detected spectrum in the region of interest (ROI) in liver-injured mouse is presented in Supplementary Figure 32a, which proves the discrimination ability of the probe over background tissue. Figure 3a shows the cross-sectional images reconstructed from

the MSOT signals from the mice at different time points upon injection of $C^1X$-$OR^1$. The upper panel of the figure shows the color images representing biodistribution of the activated probe $C^1X$-OH via spectral unmixing, while the lower panel presents the overlay of $C^1X$-OH's image with a single-wavelength (at 800 nm) image as an anatomical reference in which the spinal cord, thoracic aorta, and liver are labeled. For the mice treated with APAP, no $C^1X$-OH's signal can be observed before injection of the probe; while upon probe injection, the multispectrally resolved signal of $C^1X$-OH increases gradually within 30 min. By referring to a cryosection image of a male mouse (Fig. 3c), we could conclude that the signal is in the liver area. After 30 min upon probe injection, the MSOT signal gradually decreased over time, and after much longer time (24 h) upon probe injection, no MSOT signal could be observed (Supplementary Figure 28). This is due to the clearance and metabolic process of the dye in the mouse. Moreover, the MSOT images at different cross sections for an APAP-treated mouse are given in Supplementary Figure 29 to reflect the optoacoustic signal intensities at different cross sections. Figure 3b shows the mean signal intensity in the ROI over time for mice treated with the same procedure. From this figure, the signal increase and decay in liver area can be more clearly evaluated. Moreover, we recorded the MSOT images for the mice at different time periods post injection of 300 mg kg$^{-1}$ of APAP, and the typical images are shown in Supplementary Figure 30. After longer time post injection, the MSOT signal extended to larger area; and after 18 h the signal spread to almost all the liver region, suggesting the liver damage became severe at a longer time after APAP overdose.

MSOT technique is capable of recording multiple two-dimensional tomographic images and rendering them as 3D volume or as orthogonal maximal intensity projection (MIP) images, which serves as a common alternative to the full three-dimensional approach. In this study, we obtained orthogonal MIP images through z-stack rendering. Figure 3d reveals the z-stack MIP images for the mice with APAP treatment (300 mg kg$^{-1}$) before and 30 min after injection the liposomal probe. Upon treatment with overdosed APAP, the overall picture reflecting the injured liver can be clearly visualized, and one can find the volume of the injured liver is quite large. On the other hand, the response of the probe $C^1X$-$OR^1$ towards the treatment of APAP of varied doses is presented in Fig. 3e, f and Supplementary Figure 31, which indicates that treating a mouse with higher APAP dose causes more severe liver damage, and therefore stronger $C^1X$-OH signal in its liver region. In addition, to further prove our probe's capability to image drug-induced liver injury, we used another protocol, overdose with TNF-α in conjunction with D-galactosamine (TNF-α/D-GAL)[62], to induce the liver injury, and the results are presented in Supplementary Figure 33. Similar to APAP treatment, the treatment of TNF-α/D-GAL also caused significant increase in MSOT signal intensity in liver. In addition, we performed western blot analysis and H&E staining to prove the TNF-α/D-GAL induced hepatic cell apoptosis. As shown in Supplementary Figure 34, the level of cleaved caspase-3, an indicator of hepatic cell apoptosis, gradually increased after TNF-α/D-GAL treatment. The H&E staining further proves the hepatic cell death after TNF-α/D-GAL treatment.

The cross-sectional and z-stack MIP images also suggest that, the MSOT signal starts in left lobe at the earlier stage (6 and 12 h for APAP injection) of liver injury, and then spreads to other liver regions. Previous researches indicated that, specific lobes of the liver may be more sensitive to specific toxic agents[63,64]. To confirm the MSOT observation, we performed western blot analyses to reveal the level of phosphorylated JNK (P-JNK) in liver tissue lysis harvested from different lobes at 12 h post injection of APAP. P-JNK is over-expressed in the liver tissue of

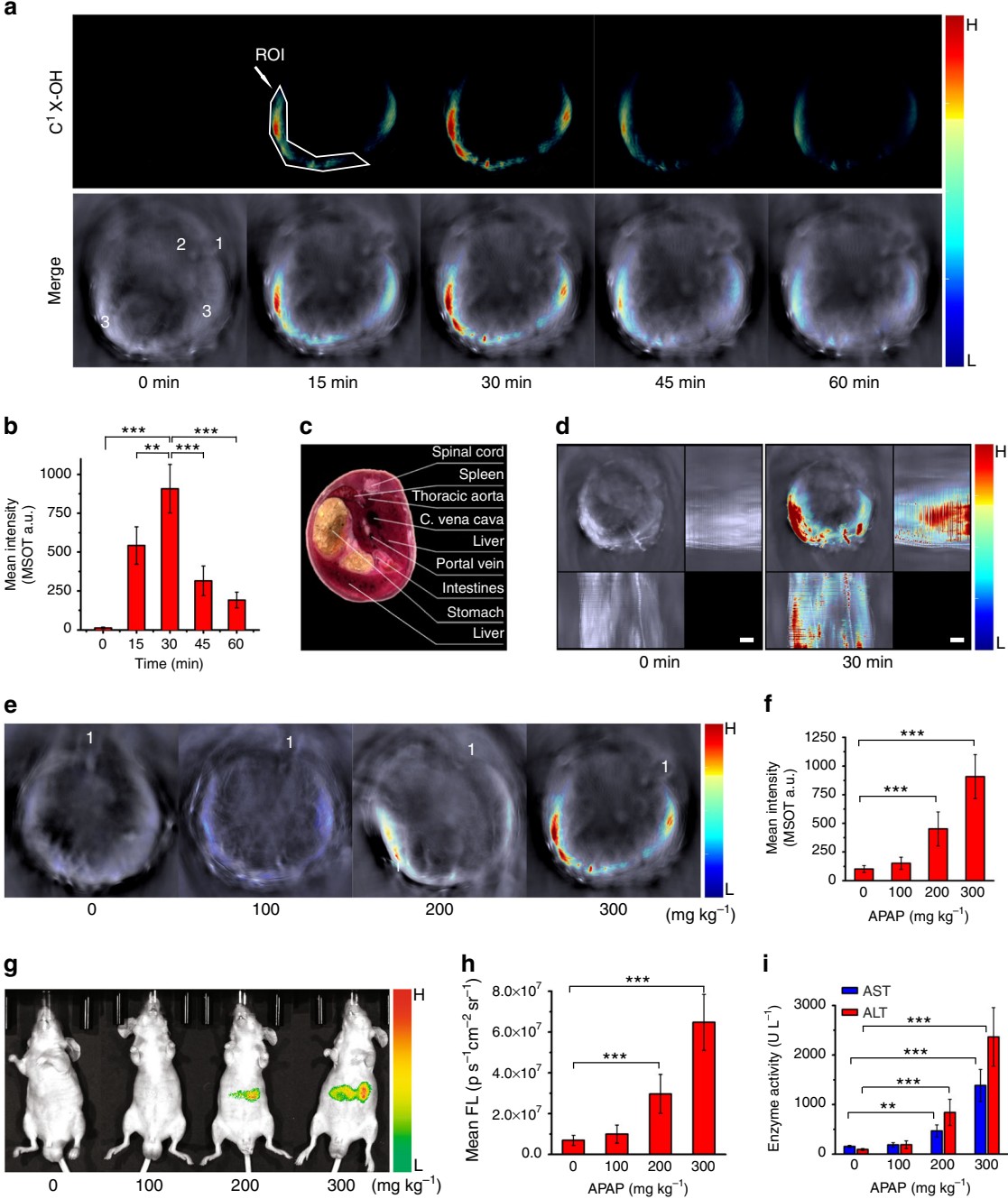

**Fig. 3** Imaging drug-induced liver injury by using $C^1X$-$OR^1$. **a** Representative cross-sectional MSOT images of a mouse at varied time points upon injection of $C^1X$-$OR^1$. The mouse was pre-treated with 300 mg kg$^{-1}$ of APAP 12 h in advance. Upper panel: multispectral resolved signal for $C^1X$-OH (activated probe). Lower panel: overlay of $C^1X$-OH's signal with the grayscale single-wavelength (800 nm) background image. Organ labeling: 1. spinal cord; 2. aorta; 3. liver. The position of spinal cord indicates the mouse lay on its chest with a certain tilt. **b** Mean optoacoustic intensities at ROI in liver area for the APAP-treated mice at varied time points upon liposomal $C^1X$-$OR^1$ injection ($n = 6$ per group). **c** A cryosection image of a male mouse with the cross section's location comparable to those shown in **a**. **d** Representative z-stack orthogonal maximal intensity projection (MIP) images for the mice pretreated with 300 mg kg$^{-1}$ APAP before and 30 min after injection of the liposomal probe. **e** Typical cross-sectional MSOT images (with background) of the mice pretreated with varied dose of APAP at 30 min upon probe injection. The images for the probe without background are presented in Supplementary Figure 31a. The spinal cords were labeled with 1 to reflect the mice's lying position in the chamber. **f** Mean optoacoustic intensities at ROI in liver area for the mice pretreated with varied dose of APAP ($n = 6$ per group). **g** Representative fluorescent images revealing biodistribution of the activated probe for the mice pretreated with varied dose of APAP and injected with the probe. The mice lay on their back during imaging, thus the distribution of fluorescent signal looks different from that of MSOT. **h** Mean fluorescent intensities at ROI of the mice pretreated with varied dose of APAP ($n = 9$ per group). **i** Serum levels of AST and ALT for the mice at 12 h upon treatment with varied dose of APAP ($n = 9$ per group). The net $C^1X$-$OR^1$ dosage for all imaging is 3.7 mg kg$^{-1}$. Columns represent means ± SD. The $p$-values (**$p < 0.01$, ***$p < 0.001$) were determined using two-sided Student's $t$-test

APAP-overdosed mice, and can be utilized to evaluate the severity of liver injury[65]. The western blot analyses shown in Supplementary Figure 35a support our MSOT observation that after a certain time upon APAP injection, the left lobe suffered more serious damage compared to other lobes. These results also indicate that, the higher dose of APAP results in higher intensity of MSOT signal and the higher level of P-JNK, suggesting the MSOT intensity is directly related to severity of the liver injury.

The fluorescent imaging for the mice upon APAP treatment was also performed. We can see from Fig. 3g that, upon APAP treatment, the mice exhibit conspicuous fluorescent signal at abdominal area; and APAP treatment with higher dose causes stronger fluorescence (Fig. 3g, h). Since $C^1X$-$OR^1$ is a dual-mode probe for detecting hepatic ALP, this result provides additional evidence that the increase in optoacoustic signal is due to the rise of hepatic ALP activity.

To verify the relationship between APAP treatment and liver injury, we measured the activity of serum alanine aminotransferase (ALT), aspartate aminotransferase (AST), and ALP by using Elisa kits. They are commonly measured clinically as markers for liver dysfunctions. Figure 3i and Supplementary Figure 31c reveals that APAP treatment causes remarkable increase in activity of the serum enzymes, proving that there is direct connection between liver dysfunction and APAP treatment. Moreover, the H&E staining histological sections from liver of the mice before and after APAP treatment are presented in Supplementary Figure 31b, which provides further evidence that the APAP-treated mice we used in the imaging experiments suffered hepatic injury. All these results indicate that, the elevation of hepatic phosphatase level as a result of liver injury can be detected by MSOT in a temporal and spatial manner.

To explore the probe's capability of tracking the rehabilitation of liver during therapeutic process, we used N-acetylcysteine (a FDA-approved antidote for APAP overdose) to feed the liver-injured mice every day and used liposomal $C^1X$-$OR^1$ (net $C^1X$-$OR^1$ dose: 3.7 mg kg$^{-1}$ in mice) to monitor the changes in optoacoustic signals, and the results are shown in Fig. 4a, b. During the medical treatment, the optoacoustic signals gradually decreased in liver area; and after six days almost no optoacoustic signal could be observed. In addition, from Fig. 4c which displays the MIP images for the mice after 0, 2 and 6 days of medical treatment, we can observe the overall liver injury status during its rehabilitation process. Furthermore, as shown in Fig. 4d, e, the fluorescent imaging of the rehabilitating mice, gives a similar result as that of optoacoustic experiments. During the medical treatment, the fluorescent intensity at liver area of the mice decreased steadily, indicating the declined hepatic ALP level in liver area during rehabilitation. To prove the relevance between the decrease in hepatic phosphatase and the liver rehabilitation, the serum levels of ALT, AST, and ALP have also been determined by using Elisa kits. As shown in Fig. 4f and Supplementary Figure 36, it is found that the levels of serum biomarkers decreased upon N-acetylcysteine treatment, confirming the liver rehabilitation after the therapy. As displayed in Fig. 4g, the microscopic images of hepatic tissue sections (H&E staining) before and after rehabilitation also prove the relevance. As another measure of liver injury, the numbers of apoptotic hepatic cells before and after N-acetylcysteine treatment were identified by TdT-mediated dUTP nick end labeling (TUNEL) staining (Fig. 4h). Much more TUNEL-positive cells were found in the liver of untreated mice than in that of N-acetylcysteine-treated mice. To investigate the rehabilitation of different liver lobes, we performed western blot analysis to reveal the P-JNK levels after N-acetylcysteine treatment for varied times. As shown in Supplementary Figure 35b, after two days of treatment, the P-

JNK level for the four liver lobes all decreased significantly, and the P-JNK levels for all lobes become very low after six days of treatment. This result is consistent with our MSOT observations shown in Fig. 4a, c.

**Detecting tumor metastasis in abdomina and lymphatic node**. Most malignant cancers tend to metastasize (spread) from a primary site to secondary site(s) within the host's body. At early stage, the metastasized tumors cannot be easily detected and located because they are small in size and deep in body[66]. In this study, by using the MSOT 3D rendering we tried to locate the metastasized tumors in abdominal cavity, which is one of the common sites where several cancers (such as ovary, colon and pancreas) spread. To verify $C^2X$-$OR^2$'s capability to image the ovarian tumor, we subcutaneously injected ovarian cells (OVCAR3 cells) at left axillary region of a mouse and then $C^2X$-$OR^2$ was intratumorally injected after a certain time (with the $C^2X$-$OR^2$ dosage of 0.32 mg kg$^{-1}$), and we observed optoacoustic and fluorescent signals, as shown in Supplementary Figure 37. The results indicate that, both the MSOT and fluorescent modalities can image the xenograft tumor as the result of ovarian cell injection. To mimic metastasis of ovarian cancer to abdominal cavity, we injected ovarian cancer cells into the abdominal cavity of female nude mice, and several weeks later injected the probe $C^2X$-$OR^2$ (at the dosage of 6.4 mg kg$^{-1}$) via intraperitoneal injection for optoacoustic imaging. The over-expressed β-galactosidase in tumor tissue in abdominal cavity[67,68] can activate the probe, and by recording optoacoustic signals of the activated probe ($C^2X$-OH) from multiple cross-sectional slices, we obtained z-stack MIP images for the mice 0, 3 or 6 weeks upon injection with ovarian cancer cells. An example of detected spectrum in metastatic tumor is presented in Supplementary Figure 32b. As we can see from Fig. 5a, at 3 weeks post injection, 2 small spots of optoacoustic signal can be observed in abdominal cavity; while after 6 weeks more and larger spots can be visualized, indicating the gradual propagation of the metastasized ovarian cancer in abdominal cavity. More importantly, the MIP images at each time point are like a three-view diagram, and one can accurately pinpoint the location of the metastasized tumors in abdominal cavity of the mice from the images. In this study, the small and deeply-located metastasized tumors in abdominal cavity could not be clearly observed fluorescently. After the mice were sacrificed and dissected, the tumors in abdominal cavity can be visualized fluorescently (Fig. 5b) or by naked eyes (Fig. 5c); and the locations of the tumors are found in good accordance with the corresponding z-stack MIP images, further proving MSOT's capability of locating the metastasized tumors. Figure 5d shows the Elisa assays of the serum CA125 (cancer antigen 125) level for the mice injected with OVCAR3 ovarian cancer cells for varied time periods. CA125 is found on the surface of many ovarian cancer cells, and usually used as an ovarian cancer marker[69]. The elevation of serum CA125, as shown in Fig. 5d, suggests that the tumors seeded in abdominal cavity were metastasized ovarian tumors. This proves that the probe $C^2X$-$OR^2$ is capable of detecting and positioning metastasized ovarian tumor through MSOT imaging.

On the other hand, lymphatic metastasis is a major mechanism for the spread of cancer[70]. In fact, many epithelial cancers first develop metastatic growth in lymphatic system by spreading via lymphatic vessels to their draining lymphatic nodes (LNs) from primary tumor, and ultimately lead to lymph node metastasis. Thus, detecting metastases within the lymphatic system, especially in sentinel LNs (the first lymphatic node into which a tumor drains) is important for evaluating a patient's prognosis and for selecting therapies. In this study, we employed MSOT and

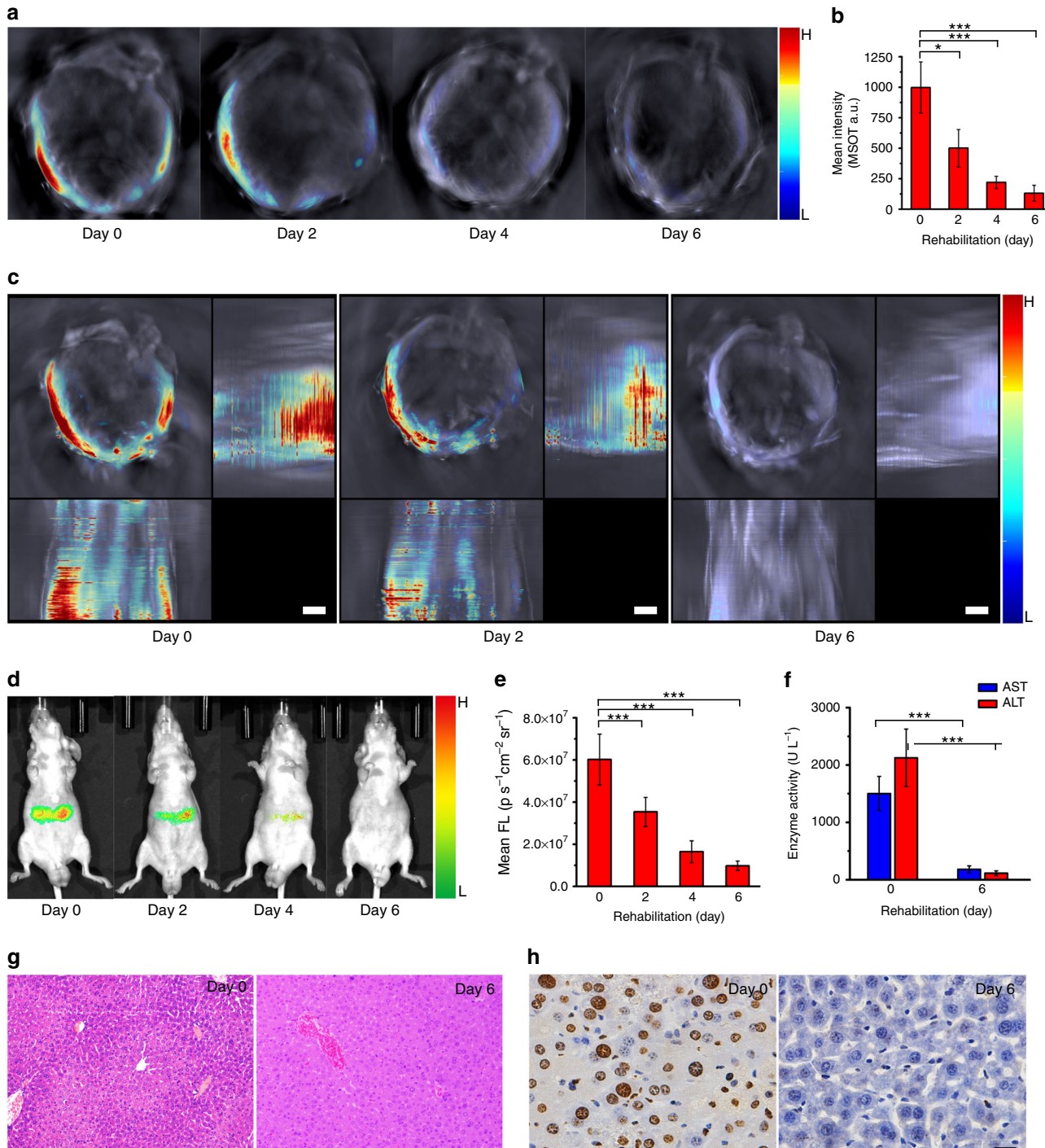

**Fig. 4** Rehabilitation of liver-injured mice revealed by imaging $C^1X$-OH level. **a**, Typical cross-sectional MSOT images for the liver-injured mice upon being orally treated with N-acetylcysteine 150 mg kg$^{-1}$ for varied days. **b**, Mean MSOT intensity in a ROI for the liver-injured mice upon N-acetylcysteine treatment for varied days ($n = 6$ per group). **c**, Representative z-stack orthogonal MIP images for the mice upon treatment for 0, 2 and 6 days (scale bar: 3 mm). **d**, Representative fluorescent images for the mice upon N-acetylcysteine treatment for varied days. **e**, Mean fluorescent intensity in a ROI for the mice upon N-acetylcysteine treatment for varied days ($n = 9$ per group). **f**, Serum level of AST and ALT for the mice upon N-acetylcysteine treatment for 0 and 6 days ($n = 9$ per group). **g**, Representative H&E staining of liver sections for the mice upon N-acetylcysteine treatment for 0 and 6 days (scale bar: 100 μm). **h**, Representative TUNEL staining of liver sections for the mice upon N-acetylcysteine treatment for 0 and 6 days (scale bar: 20 μm). The administered dose of $C^1X$-$OR^1$ was 3.7 mg kg$^{-1}$ in mice. Columns represent means ± SD. The p-values (*$p < 0.05$, ***$p < 0.001$) were determined using two-sided Student's t-test

fluorescent imaging to detect the metastasis from a primary tumor to a lymphatic node via lymphatic vessel, and results are presented in Fig. 5e–g and Supplementary Figure 38. To establish a primary tumor in mice, we injected another ovarian cancer cell line (SKOV3, a galactosidase over-expressed cell line) into the right hind footpad of BALB/c nude mice, and after varied time, the probe $C^2X$-$OR^2$ was injected into the footpad tumor (at the dose of 0.64 mg kg$^{-1}$). As revealed in the cross-sectional MSOT

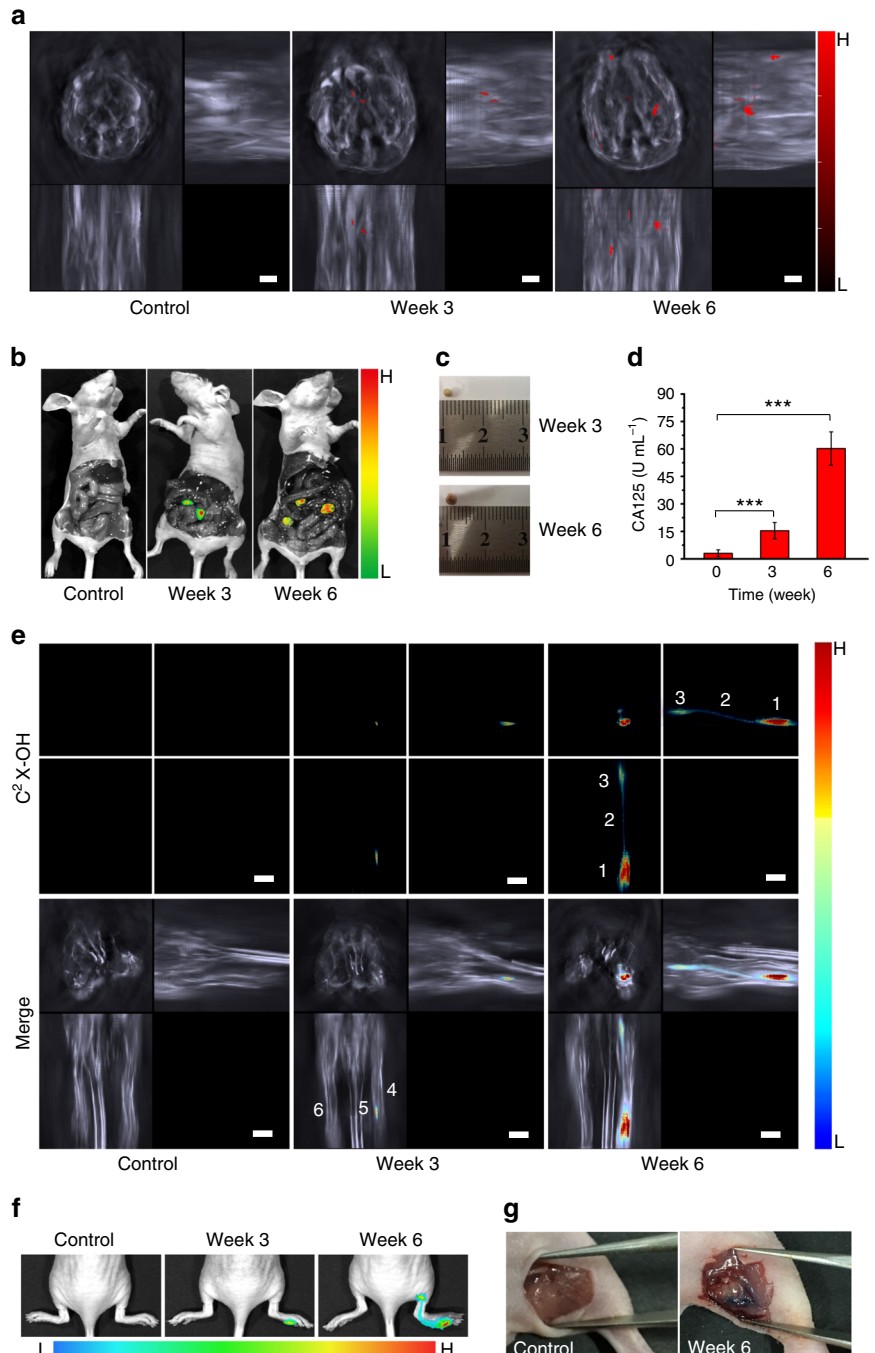

**Fig. 5** Detecting and positioning metastatic tumors by using probe $C^2X$-$OR^2$. **a** Representative z-stack orthogonal MIP images of the control group and the mice at 3 or 6 weeks upon injection of ovarian cancer cells (OVCAR3) in abdominal cavity. The images are represented by the volumetric signals of anatomical information in grayscale combined with the activated probe's signal overlaid in red. **b** Representative fluorescent images for the dissected control and the mice 3 or 6 weeks upon injection of cancer cells. The mice were the same as the corresponding ones shown in (**a**). **c** Picture of a typical tumor collected at 3 or 6 weeks after cancer cell injection. **d** Serum CA125 level for the mice at 0 (control), 3 or 6 weeks after cancer cell injection ($n = 9$ per group). **e** Representative z-stack orthogonal MIP images (for lymphatic metastasis) for the control and the mice at 3 or 6 weeks upon injection of SKOV3 at right hind footpad. Labeling: 1. Primary tumor; 2. Lymph vessel; 3. Sentinel lymph node; 4. Right hind leg; 5. Tail; 6. Left hind leg. The legs were in stretched state during imaging. **f** Representative fluorescent images for the control and the mice at 3 or 6 weeks upon injection of SKOV3 cells. **g** Pictures of the control and the mice after 6 weeks post-injection of SKOV3 cells. Skin on popliteal position was removed after euthanasia. The administered dose of $C^2X$-$OR^2$ was 6.4 mg kg$^{-1}$ for peritoneal injection and 0.64 mg kg$^{-1}$ for footpad injection. Scale bar: 3 mm. Columns represent means ± SD. The p-values (***$p < 0.001$) were determined using two-sided Student's t-test

images (Supplementary Figure 38), no MSOT signal of activated probe could be detected in the control group (the mice without cancer cell injection). In contrast, after 3 weeks post injection of cancer cells, weak signal at the primary tumor in footpad is visible; while after 6 weeks, both the primary tumor and the sentinel lymph node (popliteal lymph node) where tumor cells invade can be clearly observed. Moreover, we can map the whole metastatic route from primary tumor to sentinel lymph node by

rendering the MIP images, as shown in Fig. 5e. The fluorescent imaging (Fig. 5f) also supports our MSOT results. Figure 5g displays the images for euthanized mice (the control and the mouse 6 weeks after SKOV3 cell injection), and we can see the tumor has spread to the popliteal position in the SKOV3 cell treated mouse. In addition, we have utilized immunohistochemical (IHC) analysis of CD206 to verify the lymphatic metastasis, as shown in Supplementary Figure 39.

## Discussion

The activatable optoacoustic imaging relies on the substantial change in the absorption band of the probes. In this study, we have developed two xanthene-based activatable optoacoustic probes ($C^1X$-$OR^1$ and $C^2X$-$OR^2$) for respectively imaging liver injury and ovarian cancer metastasis by manipulating the electron-donating strength of the chromophores in the probes. The chromophores are push-pull dyes (also known as D-π-A dyes), where the electron donating moiety (D) and electron-accepting (A) are conjugated through a π-bridge. The subsequent attachment of the recognition moieties onto the donor side of the chromophores weakens electron-donating capability, thereby resulting in the blue-shift of absorption; while upon responding to the corresponding biomarkers, the cleavage of the recognition moieties transforms the weak electron-donating group into a strong one, which results in the red-shift of absorption and eventually achieves the significant changes in optoacoustic signaling. On the other hand, the push-pull structure is also beneficial for achieving intense absorption in the NIR region, and the two activated probes all exhibit relatively high extinction coefficients (higher than $5 \times 10^4 \, M^{-1} cm^{-1}$ at corresponding wavelengths), which may not be comparable to that of some metal and carbon nanoparticles, but is rather high among molecular fluorophores and enough for optoacoustic imaging.

By comparing the MSOT and fluorescence imaging shown in Figs. 3, 4, 5, we found that the MSOT imaging do exhibit some advantages. First of all, the ultrasound waves scatter less than the photons in animal tissues, they can maintain resolution and determine origin of signals at depth. Moreover, the z-stack MIP images derived from multiple cross-sectional signals can be utilized to precisely locate the disease foci, which is more advantageous than the projection-based planar images obtained from fluorescence imaging.

In this study, we developed two optoacoustic probes for detection and imaging of liver injury and tumor metastasis. The probes may find multiple applications in preclinical small animal research. For example, the probe $C^1X$-$OR^1$ may be applied in pharmaceutical industry to serve as a convenient and cost-effective system to evaluate the drug-induced liver injury, which is of great importance since drug-induced liver injury is the most common reason cited for withdrawal of an approved drug and the major reason for termination of a drug under development. On the other hand, ovarian cancer is a highly metastatic cancer, and the probe $C^2X$-$OR^2$ could be utilized to detect and track the evolution of the metastatic tumors of ovarian cancer in animal models, which may provide an in-depth understanding on the process of the tumor metastasis.

**Outlook**. This study not only serves as the proof of concept for designing molecular activatable optoacoustic contrast agents for disease diagnosis, but also demonstrated great potential of MSOT for accurately visualizing the location, dynamics, and rehabilitation of diseases with the help of activatable contrast agents. Moreover, a great number of existing NIR chromophores can be exploited this way to construct other optoacoustic probes for diagnosis and location of many other diseases.

## Methods

**Optical responses of probes in solution**. $C^1X$-$OR^1$ (or $C^2X$-$OR^2$) were dissolved in DMSO and then diluting it with TRIS (or PBS) solution under stirring for optical spectra measurement (final concentration of the probes: 5 μM). The absorbance, optoacoustic and fluorescence changes of $C^1X$-$OR^1$ upon the addition of varied amounts of ALP in 10 mM TRIS (pH = 8.0) containing 5% DMSO were recorded after 30 min of mixing. For time-course experiments, sensitivity and selectivity experiments, ALP or/and other enzymes were added into the probe solution [10 mM TRIS (pH = 8.0) containing 5% DMSO in the final test solutions]. Similarly, optical responses of $C^2X$-$OR^2$ to β-galactosidase were conducted in 10 mM PBS solution (pH = 7.4) [containing 5% DMSO in the final test solutions]. Optoacoustic signal intensity of $C^1X$-$OR^1$ and $C^2X$-$OR^2$ were recorded with an excitation wavelength of 684 nm and 703 nm respectively. Fluorescence spectra for both $C^1X$-$OR^1$ and $C^2X$-$OR^2$ were recorded with the excitation at 680 nm.

**In vitro cytotoxicity studies**. The viabilities of the four cell lines (L929, HepG2, OVCAR3, and SHIN3 cells) that were exposed to the probe were assessed by MTT assay. The cells were cultured in medium added with 10% FBS, and then seeded in 96-well plates at the cell population of about 5000 cells/well. After a 24-h period of incubation at 37 °C, the cells were washed with pre-warmed PBS solution, and then the PBS was substituted with fresh culture medium containing the probe (concentration: 0, 1, 5, 10, 50 μM; containing 0.2% v/v DMSO), after that the cells were subject to incubation for 24 h. Afterwards, the wells were washed with PBS and incubated for another 2 h with medium containing 0.5 mg mL$^{-1}$ MTT. Then, upon discarding the culture medium, 150 mL of DMSO was added to dissolve the precipitates and the absorbance was later measured with a Thermo MK3 ELISA reader at 570 nm. In the assays, for each concentration, three independent experiments were performed; and for each independent experiment, the assays were performed in eight replicates. Finally the statistical mean and standard deviation were employed to estimate cell viability.

**Activation of probes measured by flow cytometry**. The activation of $C^1X$-$OR^1$ or $C^2X$-$OR^2$ was measured by flow cytometry. HepG2 cells were seeded onto six-well plates at $2 \times 10^5$ cells per milliliter and allowed to culture for 24 h before treatment. Cells were exposed to $C^1X$-$OR^1$ or $C^2X$-$OR^2$ (5 μM) and incubated for 0.5 h and 1 h respectively. The treated cells were washed, trypsinized and centrifuged. The cells were resuspended in PBS, and approximate 10,000 cells were analyzed on a BD CS6 flow cytometer equipped with a 640 nm argon laser. The signals were collected in FL4 channels. Similarly, OVCAR3 cells were used for activation of $C^2X$-$OR^2$, and the experimental operation was the same.

**Cell imaging**. L929 and HepG2 cells were passed on polylysine-coated cell culture glass slides inside the 30-mm glass culture dishes and allowed to adhere. After 1 day, the cells were washed with culture medium, incubated in culture medium containing $C^1X$-$OR^1$ (5 μM) at 37 °C under 5% CO$_2$ for 1 h. Some HepG2 cells were incubated in culture medium containing inhibitor Na$_3$VO$_4$ (10 mM) for 30 min, and then treated with $C^1X$-$OR^1$ (5 μM) for 1 h. The control sample was incubated in culture medium only. After that, the culture dishes were washed with PBS for three times, and then subjected to fluorescence imaging. Similarly, L929, OVCAR3 and SHIN3 cells were used for cell imaging of $C^2X$-$OR^2$ (5 μM) and D-galactose (1 mM) was used as an inhibitor of β-galactosidase in OVCAR3 and SHIN3 cell imaging. The experimental operation was the same as that for cell imaging of $C^1X$-$OR^1$.

**Animal experiments**. All experiments involving animals were performed in Laboratory Animal Center of South China Agricultural University and maintained under standard conditions. All the experimental protocols have been approved by the Animal Ethics Committee of South China Agricultural University in accordance with the guidelines for the care and use of laboratory animals, as well as in compliance with the regulations on the management of laboratory animals of China and the Regulations on the Administration of Laboratory Animals of Guangdong Province. Except for the gender, mice were allocated to different groups with randomization. Blinding was not applicable. Male BALB/c nude mice (7–8 weeks old and weighing 16–20 g) for drug-induced liver injury model and female BALB/c nude mice (4 weeks old and weighing 10–15 g) for the metastatic tumor model were provided by Guangdong Medical Laboratory Animal Center (GDMLAC). As for experiments involving mouse models, sample sizes were chosen on the basis of previous studies. Mice were housed in sterile cages (three mice per cage with bedding material of sterilized wood chips) with laminar airflow hoods in a specific pathogen-free room with 12 h light/12 h dark cycle and fed autoclaved chow and water ad libitum. Before dissection operation, the mice were humanly euthanized via exposure to carbon dioxide gas in a rising concentration.

**Serum biochemistry and tissue histopathology evaluation**. For serum biochemistry test, the serum was obtained and the indexes including aminotransferase (ALT), aspartate aminotransferase (AST), ALP, and CA125 were measured by Elisa kits. For each group, nine mice were tested. For histological study, organs or tissues were excised and then flushed with sterile PBS solution, and placed in labeled glass jars full of formalin solution for paraffin embedding. Then the paraffin

sections of the tissues were made for H&E (hematoxylin and eosin) staining and imaging. The tissue histopathology was observed under a microscope.

**Mice model of drug-induced liver injury and rehabilitation**. Male mice were randomly divided into groups with six or nine mice per group, balancing sufficient replication of results with a reduction in animal number. Groups of male BALB/c nude mice (7–8 week old) were randomly selected for the following treatments. For APAP-induced liver injury, the animals were treated with varied dosage of APAP (concentration: 15 mg mL$^{-1}$, in sterilized saline solutions) via intraperitoneal administration. For TNF-α/D-galactosamine induced liver injury, D-galactosamine (D-GAL concentration: 20 mg mL$^{-1}$, dosage: 700 mg kg$^{-1}$) was intraperitoneally injected followed by the i.v. injection of recombinant human tumor necrosis factor-α (rhTNF-α, concentration: 2 μg mL$^{-1}$, dosage: 20 μg kg$^{-1}$) 5 min later, and the imaging was performed after varied time periods. For liver rehabilitation models, animals were administered with 150 mg kg$^{-1}$ of N-acetylcysteine (concentration: 7.5 mg mL$^{-1}$, in water solution) via oral gavage every day after intraperitoneally with 300 mg kg$^{-1}$ of APAP (concentration: 15 mg mL$^{-1}$, in sterilized saline solution).

**Peritoneal metastases of ovarian cancer in mice model**. Female mice were randomly divided into groups with six or nine mice per group, balancing sufficient replication of results with a reduction in animal number. Briefly, $2.5 \times 10^6$ OVCAR3 cells suspended in 250 μL of PBS (pH 7.4) were intraperitoneally injected into female BALB/c nude mice (4-week-old). Experiments with tumor-bearing mice were performed 3 or 6 week for the OVCAR3 model. For the subcutaneous tumor-bearing mice model (control), $2.5 \times 10^6$ OVCAR3 cells were inoculated s.c. into the left oxter of nude mice and incubated for 3 weeks before imaging.

**Mice model of lymph node metastasis**. Female mice were randomly divided into groups with six mice per group, balancing sufficient replication of results with a reduction in animal number. $2 \times 10^6$ SKOV3 cells suspended in 20 μL of PBS (pH 7.4) were injected into the right hind footpads of female BALB/c nude mice (4-week-old). Experiments with tumor-bearing mice were performed at 3 or 6 week post injection of SKOV3 cells.

**Multispectral optoacoustic tomography imaging**. All in-vitro phantom and in-vivo mouse optoacoustic imaging experiments were performed on a multispectral optoacoustic tomographic imaging system (inVision 128, iThera Medical GmbH). For phantom experiments, the control (PBS or TRIS solution) or the test solution was fully filled in a commercial Wilmad NMR tube respectively and then fixed on the holder of the instrument. For in vivo MSOT studies of drug-induced hepato-toxicity model, the mouse was anesthetized by 1% isoflurane delivered via a nose cone for the duration of the experiments. A catheter was then inserted into the tail vein and the mice were placed in the prone position in a water bath maintained at 34 °C, and anesthesia and oxygen are supplied through a breathing mask. 8.75 mg kg$^{-1}$ of liposomal C$^1$X-OR$^1$ (equivalent to 3.7 mg kg$^{-1}$ of net C$^1$X-OR$^1$) was injected into the mouse via tail vein. The following imaging wavelengths were selected for correspondence with the major turning points in the absorption spectra of C$_1$X-OH and hemoglobin: 680, 684, 700, 715, 730, 760, 800 (background), and 850 nm. For each wavelength, we recorded 10 individual frames. In vivo MSOT images were acquired before injection (0 min) and at different time points post injection (e.g. 15, 30, 45, and 60 min) using the MSOT system. A ROI volume consisting of transverse slices with a step size of 0.2 mm, spanning through the liver region, was selected by manual inspection of live MSOT images. For in-vivo MSOT studies of peritoneal metastases model, the mice were placed in the supine position and imaged at 2 h after intraperitoneal injection of C$^2$X-OR$^2$ at the dosage of 6.4 mg kg$^{-1}$; for in-vivo MSOT studies of lymph node metastasis model, the mice were placed in the prone position and imaged at 4 h after injection of C$^2$X-OR$^2$ at the footpad tumor with the dosage of 0.64 mg kg$^{-1}$. A ROI volume consisting of transverse slices with a step size of 0.2 mm, spanning through the abdominal cavity region, was selected by manual inspection of live MSOT images. As for the sub-cutaneous tumor-bearing mice model (serving as a control), C$^2$X-OR$^2$ (0.32 mg kg$^{-1}$, dissolved in PBS containing 5% DMSO) was injected intratumorally. The following imaging wavelengths were selected for correspondence with the major turning points in the absorption spectra of C$^2$X-OH and hemoglobin: 680, 690, 703, 715, 730, 760, 800 (background), and 850 nm. In this study, the MSOT images were reconstructed using a standard backprojection algorithm. After the images are generated, the z-stack was rendered as orthogonal MIP images. Afterwards, guided ICA spectral unmixing was utilized to separate signals from the activated probes and those from the photoabsorbing tissue elements in the body (e.g. hemoglobin). In in-vivo mouse optoacoustic imaging experiments, for each group, six mice were tested.

**Animal fluorescent (FL) imaging**. Mice were treated by tail vein injection of 8.75 mg kg$^{-1}$ liposomal C$^1$X-OR$^1$ (equivalent to 3.7 mg kg$^{-1}$ of net C$^1$X-OR$^1$) for drug-induced hepatotoxicity and its rehabilitation model or intraperitoneal injection of C$^2$X-OR$^2$ (6.4 mg kg$^{-1}$, dissolved in PBS containing 5% DMSO) for tumor peritoneal metastases model and footpad tumor injection of C$^2$X-OR$^2$ (0.64 mg kg$^{-1}$, dissolved in PBS containing 5% DMSO) for lymphatic metastasis model. Whole-body imaging was performed on an AMI small animal fluorescence imaging system (Spectral Instruments Imaging Co.) with an excitation filter of 675 nm and an emission filter of 715 nm (for C$^1$X-OH) or 735 nm (for C$^2$X-OH). For each group, nine mice were tested. All imaging parameters were kept constant for the same animal model.

**Statistical analysis**. All data were expressed in this article as mean result ± standard deviation (SD). All figures shown in the present work were obtained from certain independent experiments. No data were excluded from the analysis. Data analysis was performed using GraphPad software. Differences between groups were calculated using $t$-test for normally distributed data. Reported $p$-values were two-sided and considered significant when lower than 0.05.

**Relative OA intensity**. This value was used to indicate response of the probes to corresponding biomarker of varied concentrations for optoacoustic method. It was calculated according to the equations (1) or (2):

$$\text{For } C^1X-OR^1, \text{ Relative OA intensity} = \left[ (OA_{684})_{ALP} - (OA_{684})_{control} \right] / (OA_{684})_{control} \tag{1}$$

$$\text{For } C^2X-OR^2, \text{ Relative OA intensity} = \left[ (OA_{703})_{Gal} - (OA_{703})_{control} \right] / (OA_{703})_{control} \tag{2}$$

## Data availability

The authors declare that the data supporting the findings of this study are available within the paper and its Supplementary Information.

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

## Acknowledgements

We acknowledge the financial support by NSFC (21788102, 21875069, 51673066 and 21574044) and the Natural Science Foundation of Guangdong Province (2016A030312002). We thank Dr. C. Dai, Dr. Y. Qiu, and Dr. N. Burton for their kind helps during MSOT experiments and data processing/interpretation.

## Author contributions

Y. W., F. Z. and S. W. conceived and planned the research. Y. W. designed the synthetic route and conducted the synthesis. S. H. and J. W. performed characterization and in vitro experiments. Y. W. and L. S. performed in vivo experiments. Y. W. analyzed data.

Y. W., F. Z., and S. W. wrote the manuscript with input from all other authors. F. Z. and S. W. supervised the research.

## Additional information

**Competing interests:** The authors declare no competing interests.

