## [Peer Review File · Nature Communications]

Reviewers' Comments:

Reviewer #1:

Remarks to the Author:

In this work, the authors reported two xanthene-based PAT probes (C1X-OR1 and C2X-OR2) for temporospatial photoacoustic imaging of hepatic alkaline phosphatase (or β -galactosidase) for evaluating and locating drug-induced liver injury (or metastatic tumor) in vivo. The probes responded to the disease-specific biomarkers by displaying red-shifted NIR absorption bands and thereby generate photoacoustic signals. By adopting three-dimensional photoacoustic tomography, the authors could precisely locate the focus of drug-induced liver injury in mice using C1X-OR1, as well as the metastatic tumors in abdominal cavity using C2X-OR2. This work suggests that the activatable PAT chromophores may potentially be applied for diagnosing and positioning disease focuses, especially smaller and deeper ones.

The study is original and innovative. The presented results are interesting and significant. There are some minor points listed below, which the authors might find helpful to strengthen the paper:

- 1) On Page 10, Line 179, "...no red intracellular fluorescence..." is suggested to be "...no intracellular red fluorescence...".
- 2) On Page 18, Line 293, "It is found as shown in Fig. 4f and Supplementary Fig. 28, the levels of serum biomarkers 294 decreased..." is suggested to be "As shown in Fig. 4f and Supplementary Fig. 28, it is found that the levels of serum biomarkers 294 decreased...".

ζ

Reviewer #2:

Remarks to the Author:

The paper discusses the development of activatable probes for photoacoustic imaging of liver injury and tumor metastasis. The probes are based on xanthene and respond to hepatic alkaline phosphatase, by shifting their absorption spectrum. Developing novel agents for an imaging modality is work that has merit. On the other hand, since many different agents can be generated, it is important that any of these agents presented can enable ability not otherwise possible.

The largest issue with the study performed is that of sensitivity. It is unclear which concentration is imaged in the work performed. The authors should provide a titration study of their agent in realistic conditions (i.e. blood phantom, euthanized mice or similar) in order to describe with accuracy the relation between signal detected and sensitivity. This is essential for understanding the value of the work performed. This comment is not about administered dose, but about the actual concentration imaged non-invasively in mice. This is particularly important in understanding the detection limits in metastatic disease in Fig.5.

Overall I found it awkward that the administered dose is not listed in the main manuscript but rather appears only in methods. Please prominently list the administered dose in the main text and legend of the figures 4,5.

Plotting the spectra detected would be equally useful, so as to understand the discrimination ability over background tissue not containing the agent.

Finally the discussion is not convincing. What would be the utility of the probes developed? I did not find any convincing argument on which applications these agents may prove useful.

Furthermore the particular work makes erroneous statements and confuses optical and photonic so often that requires a thorough revision:

The authors claim the PA signal depends on the absorption property of an agent (page 7); however this statement fails to acknowledge the effects of quantum yield.

The authors mix names and references. PAT is a generic term associated with general photoacoustic / optoacoustic imaging. MSOT is a spectral PA/OA technique. The authors give primarily references to PAT, but fail to reference the large volume of work on MSOT, which appears unbalanced, given that they use the MSOT technique and an MSOT system. Or reference work that is not on MSOT, as part of MSOT. They also mix names referring to MSOT as Multispectral Photoacoustic Tomography or multispectral PAT (page 12, page 13). The authors should do a much better job explaining MSOT, referencing the relevant literature on MSOT and properly refer to the technique they use.

It is also unclear why the authors refer to "optical" absorption, "optical" responses, "optical" imaging, but call MSOT "photoacoustic" and PAT. It is optical and optoacoustic imaging or photonic and photoacoustic imaging. Why mix and match? Moreover, "MSOT" does not yield "PAT" signals as prevalently listed in the paper – this is a misnomer. Either call everything "photonic" or call everything "optical". This alternation is very confusing in the text.

The technique they use is not three dimensional. Two dimensional images are obtained. Yet the authors mistakenly call it 3D.

Reviewer #3:

Remarks to the Author:

In the paper "Activatable probes for diagnosing and positioning liver injury and metastatic tumors by using photoacoustic tomography" by Yinglong Wu and colleagues the authors describe a novel methodology to monitor liver damage, generate 3D images in the liver, locate signs of damage with high resolution as well as monitor metastatic tumors. Many of the herewith shown experiments are of excellent quality and most of them are well controlled. Having said this several experiments, controls and approaches are missing to corroborate the usefulness of this novel, interesting technique. In general this study needs a native speaker to read through the paper one more time, there are several typos throughout the manuscript. In general, very often statistical analyses are omitted this should be worked on.

Specific comments:

Figure 1) The schematic representation in figure 1 is well done and straight forward. It brings together all necessary parts for non-specialists to understand the paper.

Figure 2) This Figure is clear and straight forward, although controlled in a different environment.

Figure 3) In vivo analysis is well performed and several 3D images are spectacular. However, several things could be added. A) A longer time course would be feasible. B) Why are only specific parts of the liver affected or is this a technical issue. APAP induces liver damage in all regions of the liver. Is the intensity directly correlated with severity of liver damage? This should be analyzed.

C) In addition, another model of liver damage (e.g. TNF-D-Gal) should be used to monitor liver damage and corroborate the technique.

Figure 4) Experiments on reversal of liver damage are interesting and also nice. Histology should be corroborated by IHC for cell death (e.g. Cleaved Caspase 3) and molecular analyses (Western blot for makers od cell death). It should be indicated in which regions the damage is reduced first. Which lobes? In case this technique should proof its feasibility these questions of resolution and definition should be answered.

Figure 5) The detection and positioning of metastatic tumors is very nice. This referee is missing

another tumor entity – e.g. in the liver (e.g. colon cancer cell liver metastasis by splenic injection or spontaneous tumors). This should be added to show the flexibility of the technique. Resolution of this technique is enormous.

I believe this is a very interesting functional non-invasive technology that might have the potential to change several approaches in mouse models and prepare for techniques of this technology in vivo in patients. Still several experiments need to be done before being appropriate for acceptance by Nature Communications.

Reviewers' comments:

Reviewer #1 (Expertise: Optical imaging probes, Remarks to the Author):

In this work, the authors reported two xanthene-based PAT probes (C1X-OR1 and C2X-OR2) for temporospatial photoacoustic imaging of hepatic alkaline phosphatase (or β -galactosidase) for evaluating and locating drug-induced liver injury (or metastatic tumor) in vivo. The probes responded to the disease-specific biomarkers by displaying red-shifted NIR absorption bands and thereby generate photoacoustic signals. By adopting three-dimensional photoacoustic tomography, the authors could precisely locate the focus of drug-induced liver injury in mice using C1X-OR1, as well as the metastatic tumors in abdominal cavity using C2X-OR2. This work suggests that the activatable PAT chromophores may potentially be applied for diagnosing and positioning disease focuses, especially smaller and deeper ones.

The study is original and innovative. The presented results are interesting and significant. There are some minor points listed below, which the authors might find helpful to strengthen the paper:

Our response: We thank the reviewer for the comments and pointing out some inappropriate expressions and typos. We have checked our manuscript again and corrected some inappropriate expressions and typos.

1) On Page 10, Line 179, "...no red intracellular fluorescence..." is suggested to be "...no intracellular red fluorescence...".

Our response: We have corrected this inappropriate expression on page 10 of revised manuscript.

2) On Page 18, Line 293, "It is found as shown in Fig. 4f and Supplementary Fig. 28, the levels of serum biomarkers 294 decreased..." is suggested to be "As shown in Fig. 4f and Supplementary Fig. 28, it is found that the levels of serum biomarkers 294 decreased...".

Our response: We have corrected this typo, and the revised sentence is now on page 19 of the revised manuscript.

Reviewer #2 (Expertise: Biomedical imaging, therapy, Remarks to the Author):

The paper discusses the development of activatable probes for photoacoustic imaging of liver injury and tumor metastasis. The probes are based on xanthene and respond to hepatic alkaline phosphatase, by shifting their absorption spectrum. Developing novel agents for an imaging modality is work that has merit. On the other hand, since many different agents can be generated, it is important that any of these agents presented can enable ability not otherwise possible.

- The largest issue with the study performed is that of sensitivity. It is unclear which concentration is imaged in the work performed. The authors should provide a titration study of their agent in realistic conditions (i.e. blood phantom, euthanized mice or similar) in order to describe with accuracy the relation between signal detected and sensitivity. This is essential for understanding the value of the work performed. This comment is not about administered dose, but about the actual concentration imaged non-invasively in mice. This is particularly important in understanding the detection limits in metastatic disease in Fig.5.

Our response: We thank the reviewer for the comments. We have determined the sensitivity limit of the activated probes by referring to literature reports (*IEEE Trans. Med. Imaging.* 2016, 35, 2534; *IEEE Trans. Med. Imaging.* 2014, 33, 48). The determined sensitivity limits for C₁X-OH and C₂X-OH were 2.3 μM ($\mu_a = 0.177 \text{ cm}^{-1}$) and 3.3 μM ($\mu_a = 0.180 \text{ cm}^{-1}$) respectively. The relevant discussion is given on first paragraph on page 9 of the revised manuscript and in Supplementary Figure S22 (on page s17 and s18 of revised Supplementary Information).

- Overall I found it awkward that the administered dose is not listed in the main manuscript but rather appears only in methods. Please prominently list the administered dose in the main text and legend of the figures 4, 5.

Our response: We have added the administered dose of the probes in the main text (on pages 13, 18, 22 and 23) and in the legends of Figures 3, 4 and 5.

- Plotting the spectra detected would be equally useful, so as to understand the discrimination ability over background tissue not containing the agent.

Our response: We obtained the spectra detected and gave an example for each probe as shown in Supplementary Figure S31 in the revised Supplementary Information. We have mentioned this point on pages 13 and 22 in the revised manuscript.

- Finally the discussion is not convincing. What would be the utility of the probes developed? I did not find any convincing argument on which applications these agents may prove useful.

Our response: We have added the following statement in the Discussion part (on page 25) of the revised manuscript.

In this study, we developed two optoacoustic probes for detection and imaging of liver injury and tumor metastasis. The probes may find multiple applications in preclinical small animal research. For example, the probe C₁X-OR₁ may be applied in pharmaceutical industry to serve as a convenient and cost-effective system to evaluate the drug-induced liver injury, which is of great importance since drug-induced liver injury is the most common reason cited for withdrawal of an approved drug and the major reason for termination of a drug under development. On the other hand, ovarian cancer is a highly metastatic cancer, and the probe C₂X-OR₂ could be utilized to detect and track the evolution of the metastatic tumors of ovarian cancer in animal models, which may provide in-depth understanding on the process of the tumor metastasis.

- Furthermore the particular work makes erroneous statements and confuses optical and photonic so often that requires a thorough revision:

Our response: We thank the reviewer for pointing out our erroneous and inappropriate statements. We have made thorough revisions throughout the manuscript.

- The authors claim the PA signal depends on the absorption property of an agent (page 7); however this statement fails to acknowledge the effects of quantum yield.

Our response: We have corrected this inappropriate expression by acknowledging the effect of quantum yield in the revised manuscript (in the second paragraph on page 7).

- The authors mix names and references. PAT is a generic term associated with general photoacoustic / optoacoustic imaging. MSOT is a spectral PA/OA technique. The authors give primarily references to PAT, but fail to reference the large volume of work on MSOT, which appears unbalanced, given that they use the MSOT technique and an MSOT system. Or reference work that is not on MSOT, as part of MSOT. They also mix names referring to MSOT as Multispectral Photoacoustic Tomography or multispectral PAT (page 12, page 13). The authors should do a much better job explaining MSOT, referencing the relevant literature on MSOT and properly refer to the technique they use.

Our response: In this revised manuscript, we have corrected the confusion in names, and also added a detailed description on MSOT in the Introduction section (on page 2 of the revised manuscript). We also modified some MSOT-related phrases in the revised manuscript and added/changed some MSOT-related references (on pages 33-36).

- It is also unclear why the authors refer to “optical” absorption, “optical” responses, “optical” imaging, but call MSOT “photoacoustic” and PAT. It is optical and optoacoustic imaging or photonic and photoacoustic imaging. Why mix and match? Moreover, “MSOT” does not yield “PAT” signals as prevalently listed in the paper – this is a misnomer. Either call everything “photonic” or call everything “optical”. This alternation is very confusing in the text.

Our response: we thank the reviewer for this suggestion. We have used the terms “optical”, “optoacoustic”, “OA” and “MSOT” throughout the revised manuscript.

- The technique they use is not three dimensional. Two dimensional images are obtained. Yet the authors mistakenly call it 3D.

Our response: We thank the reviewer for pointing out our mistake. The technique (*J. Biomed. Opt.* 2014, **19**, 36021) we used is actually the two-dimensional imaging. The MSOT technique is capable of achieving volumetric (full three-dimensional) imaging. While in many cases, after multiple 2D cross-sectional images were generated, the z-stack can be rendered as a 3D volume or as orthogonal maximal intensity projection (MIP) images. This is a common alternative to the full three-dimensional approach. In this study, we showed orthogonal MIP

images, and these MIP images can reflect 3D information and can help to locate the disease focus.

In this revised manuscript, we have mentioned this point (in second paragraph, on page 14). We also have rephrased our manuscript and mainly used the phrase “z-stack orthogonal MIP images”.

Reviewer #3 (Expertise: Liver injury, metastasis models, Remarks to the Author):

In the paper “Activatable probes for diagnosing and positioning liver injury and metastatic tumors by using photoacoustic tomography” by Yinglong Wu and colleagues the authors describe a novel methodology to monitor liver damage, generate 3D images in the liver, locate signs of damage with high resolution as well as monitor metastatic tumors. Many of the herewith shown experiments are of excellent quality and most of them are well controlled. Having said this several experiments, controls and approaches are missing to corroborate the usefulness of this novel, interesting technique. In general this study needs a native speaker to read through the paper one more time, there are several typos throughout the manuscript. In general, very often statistical analyses are omitted, this should be worked on.

Our response: we thank the reviewer for the comments. We have checked our manuscript and we have revised some inappropriate expressions and corrected some typos. We also added statistical analyses for our data.

Specific comments:

Figure 1) The schematic representation in figure 1 is well done and straight forward. It brings together all necessary parts for non-specialists to understand the paper.

Figure 2) This Figure is clear and straight forward, although controlled in a different environment.

Figure 3) In vivo analysis is well performed and several 3D images are spectacular. However, several things could be added. A) A longer time course would be feasible.

Our response: We have recorded the cross-sectional MSOT images for the mice (overdosed with 300 mg/kg of APAP in advance) for longer time course (at 1 h, 12 h and 24 h)

upon probe injection, and the result is given in Supplementary Figure S27. After the probe was injected into the liver-injured mice, the MSOT signals at the liver region began to emerge and then became stronger (Fig. 3a); and after a certain time period, the signal became weaker due to the metabolic function of liver (Fig. 3a and Supplementary Figure S27), and we found after 24 h, the signal could not be observed (Supplementary Figure S27). The relevant point has been mentioned in first paragraph on page 14 of the revised manuscript.

B) Why are only specific parts of the liver affected or is this a technical issue. APAP induces liver damage in all regions of the liver. Is the intensity directly correlated with severity of liver damage? This should be analyzed.

Our response: Figure 3 suggests that, at 12 hour post-injection of APAP, the MSOT signal in the left lobe of the liver of the APAP-treated mice is stronger than that in other lobes. To further investigate the distribution of MSOT signals in the damaged liver of the APAP-treated mice, we recorded the MSOT images for the mice at 6 hour and 18 hour post-injection of APAP, and the result is added in the revised Supplementary Information of the revised manuscript (Supplementary Figure S29). These results suggest that, the MSOT signal starts in the left lobe at the early stage of liver damage, and the signal spreads to other liver region at longer time. Some previous researches have suggested that specific lobes of the liver may be more sensitive to specific toxic agents. (C. H. Frith, et al., *Toxicologic pathology*, 1981, 9, 1; T. A. Lawson, et al., *Br. J. exp. Path.* 1974, 55, 583). For example, Lawson and et al. investigated the extent (severity) of liver damage in rats treated with CCl₄ and dimethylnitrosamine (DMN) and compared the extent of damage in different lobes (*Br. J. exp. Path.* 1974, 55, 583). They found the extent of live damage was greater in the right than in the left lobes of animals treated with CCl₄, but was greater in the left lobes of animals given with DMN. In this study, we found APAP caused more severe damage to left lobe at the same time point post injection of drug.

Our results (Fig. 3e and 3f, Supplementary Fig. S34a) also suggest that, the higher dose of APAP results in higher intensity of MSOT signal and the higher level of P-JNK (a marker for APAP-induced liver injury), suggesting the MSOT intensity is directly related to severity of the liver injury.

We have added these discussions from the last paragraph on page 15 of the revised manuscript.

C) In addition, another model of liver damage (e.g. TNF-D-Gal) should be used to monitor liver damage and corroborate the technique.

Our response: We have performed liver injury experiments by overdosing the mice with TNF/D-Gal, and the cross-sectional and z-stack-rendering 3D images were recorded at 0 h, 2 h, 4 h and 8 h post-injection of TNF/D-Gal. The results are presented in Supplementary Fig. S32 and S33, which indicates that, the overdose of TNF/D-Gal can also cause the elevation of ALP and can be detected by the optoacoustic tomographic technology. The relevant discussion is presented in the first paragraph on page 15 of the revised manuscript.

Figure 4) Experiments on reversal of liver damage are interesting and also nice. Histology should be corroborated by IHC for cell death (e.g. Cleaved Caspase 3) and molecular analyses (Western blot for makers of cell death). It should be indicated in which regions the damage is reduced first. Which lobes? In case this technique should proof its feasibility these questions of resolution and definition should be answered.

Our response: We thank the reviewer for the comment and suggestions. The reports on the effect of APAP overdose on cleaved caspase-3 level are discrepant. Some studies indicate that the cleaved (active) caspase-3 increases after APAP treatment (e. g. A. Kumari, P. Kakkar, *Life Sciences* 90 (2012) 561–570); while some other reports (e. g. *Toxicol Appl Pharmacol.* 2011, 257, 449; *J. Clin. Invest.* 2012, 122, 1574) suggest that, the cleaved caspase-3 does not increase or only slightly increase after APAP overdose. Therefore, in this study, for APAP-induced liver injury, we used TUNEL analysis to evaluate the hepatocyte apoptosis. This result is presented in Figure 4h, which clearly indicates the hepatocyte apoptosis upon APAP overdose. The relevant discussion is given on page 19 of the revised manuscript.

On the other hand, previous studies (e.g. *Hepatology*, 2010, 52, 691; *Hepatology*, 2007, 45, 412) have demonstrated that reactive oxygen species (ROS) derived from APAP bioactivation directly activates JNK and cause the phosphorylation of JNK (P-JNK) through MAPK pathway. Together with ROS, activated JNK can stimulate the expression of

proapoptotic proteins and block the function of antiapoptotic proteins, leading to serious hepatotoxicity and cell apoptosis. Thus, we performed Western blot analyses with p-JNK as the marker to evaluate the APAP-induced liver injury and the subsequent recovery. The results indicate that, the liver injury starts at left lobe and the recovery starts at all lobes, which is in accordance with our MSOT observations. The results are given in Supplementary Figure S34 and the relevant discussions are presented in second paragraph on page 15 of the revised manuscript.

(By the way, many reports indicate that the level of cleaved caspase-3 increases in the mice overdosed with TNF-D-Gal. Thus, for TNF-D-Gal induced liver injury model, we used caspase-3 as the marker for cell apoptosis in Western blot analysis. The result is given in Supplementary Figure S33a, and the relevant discussion is mentioned in the first paragraph on page 15).

Figure 5) The detection and positioning of metastatic tumors is very nice. This referee is missing another tumor entity – e.g. in the liver (e.g. colon cancer cell liver metastasis by splenic injection or spontaneous tumors). This should be added to show the flexibility of the technique. Resolution of this technique is enormous.

Our response: We thank the reviewer for this suggestion. In general, colon cancer cells do not over-express galactosidase unless gene transfection is performed (C. Lengauer et al., *Proc. Natl. Acad. Sci.*, 1997, 94, 2545; K. Gu et al., *J. Am. Chem. Soc.* 2016, 138, 5334), and currently we do not have this kind of gene transfection skill. Instead, we used optoacoustic tomography to image another tumor entity, the lymphatic metastasis of ovarian cancer. Lymphatic metastasis is an important mechanism in the spread of cancers, and lymphatic vessels could serve as the highway for tumor metastasis, as lymph flow promotes dissemination of cancer cells between the primary tumor and the lymph nodes and ultimately leads to lymph node metastasis. (E. R. Pereira, D. Jones, K. Jung, T. P. Padera, *Semin. Cell Dev. Biol.*, 2015, 38, 98-105). In this study, we injected another ovarian cancer cell line (SKOV3 cell, a galactosidase over-expressed cell line) into the right hind footpad of BALB/c nude mice; and after a certain time, both the primary tumor at footpad and the sentinel lymph node where tumor cells invade are clearly observed in cross-sectional MSOT images.

Moreover, we can map the whole metastatic route from primary tumor to sentinel lymph node by recording the z-stack rendering 3D images of the metastatic route. The results are given in Figure 5e-5g in the revised manuscript and in Supplementary Fig. S37 and S38 in the revised Supplementary Information, and the relevant discussion is given on the legend of Fig. S38 (on page s34) in the revised Supplementary Information as well as in the revised manuscript (on pages 23 and 24).

Best Regards,

Sincerely yours,

Shuizhu Wu, Ph.D., Professor
College of Materials Science & Engineering
State Key Laboratory of Luminescent Materials & Devices
South China University of Technology
Guangzhou 510640, China
Tel: 86-20-22236262
e-mail: shzhwu@scut.edu.cn